# Effectiveness of Human Mobility Change in Reducing the Spread of COVID-19: Ecological Study of Kingdom of Saudi Arabia

**Mohamed Ali Alzain** [1,2,*] , **Collins Otieno Asweto** [3] , **Suleman Atique** [4] , **Najm Eldinn Elsser Elhassan** [1] , **Ahmed Kassar** [1] , **Sehar-un-Nisa Hassan** [1] , **Mohammed Ismail Humaida** [1] , **Rafeek Adeyemi Yusuf** [5] and **Adeniyi Abolaji Adeboye** [1,6]

1 Department of Public Health, College of Public Health and Health Informatics, University of Ha'il, Ha'il 81451, Saudi Arabia; n.hamedelneel@uoh.edu.sa (N.E.E.E.); a.kassar@uoh.edu.sa (A.K.); s.nisa@uoh.edu.sa (S.-u.-N.H.); m.humaida@uoh.edu.sa (M.I.H.); adeboye05@yahoo.co.uk (A.A.A.)
2 Departments of Community Medicine, Faculty of Medicine and Health Sciences, University of Dongola, Dongola, P.O. Box 47, Dongola 41111, Sudan
3 Department of Community Health, School of Nursing, University of Embu, Embu P.O. Box 6-60100, Kenya; aswetotieno@gmail.com
4 Department of Health Informatics, College of Public Health and Health Informatics, University of Ha'il, Ha'il 81451, Saudi Arabia; su.atique@uoh.edu.sa
5 Department of Management, Policy and Community Health, University of Texas Health Sciences at Houston, Houston, TX 77030, USA; rafeeky8@gmail.com
6 Department of Health Promotion and Behavioral Sciences, University of Texas Health Sciences at Houston, Houston, TX 77030, USA
* Correspondence: m.alzain@uoh.edu.sa; Tel.: +966-54-839-8414

**Abstract:** Non-pharmacological interventions including mobility restriction have been developed to curb transmission of SARS-CoV-2. We provided precise estimates of disease burden and examined the impact of mobility restriction on reducing the COVID-19 effective reproduction number in the Kingdom of Saudi Arabia. This study involved secondary analysis of open-access COVID-19 data obtained from different sources between 2 March and 26 December 2020. The dependent and main independent variables of interest were the effective reproduction number and anonymized mobility indices, respectively. Multiple linear regression was used to investigate the relationship between the community mobility change and the effective reproduction number for COVID-19. By 26 December 2020, the total number of COVID-19 cases in Saudi Arabia reached 360,690, with a cumulative incidence rate of 105.41/10,000 population. Al Jouf, Northern Border, and Jazan regions were ≥2.5 times (OR = 2.93; 95% CI: 1.29–6.64), (OR = 2.50; 95% CI: 1.08–5.81), and (OR = 2.51; 95% CI: 1.09–5.79) more likely to have a higher case fatality rate than Riyadh, the capital. Mobility changes in public and residential areas were significant predictors of the COVID-19 effective reproduction number. This study demonstrated that community mobility restrictions effectively control transmission of the COVID-19 virus.

**Keywords:** COVID-19; transmission; effective reproduction number; community mobility restriction; case fatality rate

## 1. Introduction

The coronavirus disease 2019 (COVID-19) pandemic, an ongoing global public health crisis, has engulfed the lives of 5.5 million people to date [1]. The natural history including the etiology, transmission pattern, symptomatology, treatment, and prevention of COVID-19 have been well documented [1–4]. It has significantly and negatively impacted the well-being of individuals, organizations, communities, and systems [1–4]. Non-pharmacologic and pharmacologic interventions have been developed to stem the spread of severe acute respiratory syndrome corona virus type 2 (SARS-CoV-2), the causative agent of COVID-19.

Mobility restriction is one of the non-pharmacologic interventions (NPIs) recommended and implemented worldwide [5–9]. There has been varying success of this preventive measure based on disease-related factors, geographical location, population density, the economy, existing health infrastructures, human resources, traditional and cultural norms, and the health policies of affected jurisdictions [5–9]. For instance, in a study using mobile data from nine European countries, the magnitude of mobility reduction was strongly correlated with decrease in the basic reproductive number ($R_0$) [6]. In an Omani study, mobility restrictions in the form of evening lockdowns significantly affected the course of the pandemic in the country [7]. In a United States study mobility restrictions were observed as a consequence and means of curbing the spread of SARS-CoV-2 [8]. Similarly, in Africa, a study conducted involving 26 countries reported that a reduction in public spaces mobility is an effective COVID-19 containment strategy [9].

In the Kingdom of Saudi Arabia (KSA), the Ministry of Health (MoH) reported the first case of COVID-19 infection on 2 March 2020 [10]. This was a Saudi national returning to KSA via air travel from Iran through Bahrain [11–13]. The government of Saudi Arabia immediately responded by instituting nationwide mobility restrictions including (1) internal measures—such as prolonged curfew hours, closure of recreational places, mosques, gyms, saloons, dining places, and shopping malls and (2) external measures—such as restriction of operations at all ports of entries (air, land, and sea) into and out of KSA [14,15]. Most of the activities and services offered by public and private sector organizations were now being conducted virtually [15].

The published studies from KSA related to COVID-19 mobility restriction policy focused mostly on awareness and predictors of compliance with the policy [16–20]. Quantifying the impact of mobility restriction policy on the spread and consequences of COVID-19 at the early, intermediate, and late phases of the pandemic in KSA has been challenging [11–15]. Therefore, this study was designed to provide more precise estimates of disease burden and measure the impact of mobility restriction in preventing the spread of COVID-19. To the best of our knowledge, this is the first study quantifying the impact of the mobility restriction policy on the transmission of and case fatalities associated with COVID-19 in KSA.

## 2. Materials and Methods

### 2.1. Study Design

This was an observational study involving analysis of secondary data derived from (1) KSA COVID-19 Community Mobility Report from Google [21] and (2) COVID-19 epidemiological data obtained from the KSA Ministry of Health website [22]. The COVID-19 data generated from 2 March 2020 to 26 December 2020 were used for this study.

### 2.2. Data Sources

2.2.1. Google-Derived KSA COVID-19 Community Mobility Report

COVID-19 community mobility reports were obtained from the Google website (https://www.google.com/covid19/mobility/ (accessed on 24 June 2021). These community reports are an aggregate of anonymized data collected by the Google team to provide insight into changing community mobility initiated by the policy developed to stem the transmission of COVID-19. The reports show changes in spatio-temporal mobility trends across varying community locations such as residential, workplaces, recreation centers, transit stations, groceries and retail, and healthcare (hospitals, clinics, and pharmacies) centers. Daily mobility changes for each of the locations stated above were compared to the baseline value for that day. The baseline value is the median value of relatively normal (non-widespread-COVID-19 –induced) mobility for the corresponding day of the week during the five-week period of 3 January to 6 February 2020 [21]. The mobility report depicts changes in mobility trends over several weeks, in this case, over the study period of 2 March 2020 to 26 December 2020.

### 2.2.2. KSA Ministry of Health-Derived COVID-19 Epidemiology Data

KSA is the second largest country in the Arab world with a population of over 34 million. Of this, non-citizens of Saudi Arabia represent approximately 37% [23]. Saudi Arabia has 13 administrative regions comprising Riyadh, Makkah, the Eastern, Madinah, Al Baha, Al Jouf, Northern Borders, Qassim, Ha'il, Tabuk, Aseer, Jazan, and Najran.

COVID-19 epidemiological data abstracted from this source and used in this study included daily-confirmed cases, recoveries, and mortalities related to COVID-19 [22]. The Saudi MoH reports daily numbers of confirmed COVID-19 cases based on real-time polymerase chain reaction (RT-PCR) tests obtained through nasopharyngeal swabs, which were processed, validated, and reported through a regional lab. A confirmed COVID-19 case is defined as a case with a positive real-time (RT-PCR) sample obtained through nasopharyngeal swabs [22].

### 2.3. Measures

The dependent variable of interest in this study was the case fatality rate (CFR) defined as the number of deaths due to COVID-19 divided by the number of COVID-19-confirmed cases (cumulative incidence) per 100, while the cumulative incidence was the proportion of confirmed cases divided by the total population per 100,000 for the entire KSA and each of the 13 administrative regions [24].

While the basic or initial reproduction number ($R_0$) of SARS-CoV-2 indicates a measure of susceptibility to the virus, the effective reproduction number ($R_t$) is an indication of an actual number of infected cases taking into account the number of people infected or immunized overtime during the pandemic. Hence, in monitoring the effectiveness of mobility restriction in stemming the spread of SARS-CoV-2 in KSA, the effective reproduction number ($R_t$) was used; $R_t$ has been demonstrated to be a more objective measure at a given point in time than the initial reproduction number ($R_0$) [25,26]. The $R_t$ is the mean number of secondary cases generated by a typical primary case at time t in a population. The $R_t(t_i)$ of COVID-19 was estimated using a simple mathematical formula proposed by Contreras et al. [26].

$$R_t(t_i) = (-\Delta_i S)/(\Delta_i R + \Delta_i D) \tag{1}$$

where $-\Delta_i S$ is new infections; $\Delta_i R$ is new clinical recoveries; and $\Delta_i D$ is new deaths.

This formula is more appropriate for estimating the effective reproduction number ($R_t$) directly from raw and real-time data of an evolving epidemic outbreak. The estimate has also been demonstrated to be a helpful decision parameter to assess the impact of non-pharmaceutical interventions (NPIs) in controlling COVID-19 [26].

The independent variables of interest in this study were anonymized mobility indices from each of the 13 administrative regions in KSA. These indices were derived from specific categories of locations comprising residential, workplaces, recreation centers, transit stations, groceries and retail, and healthcare (hospitals, clinics, and pharmacies) centers. The covariate of interest in this study was the effective reproduction number ($R_t$) of SARS-CoV-2. The average mobility for the specified categories of locations between 3 January and 6 February 2020 (the period before COVID-19 was declared a global pandemic) was used as the reference mobility value to determine the percent mobility change on each day during the study interval for each administrative region in KSA.

### 2.4. Data Analysis

A descriptive statistic (mean, standard deviation, maximum, and minimum value) was reported for mobility trends. The overall trends of average weekly changes in the COVID-19 effective reproduction number and changes in mobility indices in each categorized location (retail and recreation, grocery and pharmacy, parks, transit stations, workplaces, and residential) were observed for the entire study period.

Chi-square tests and logistic regression were used to determine whether there were statistically significant differences in CFR (dependent variable) between the 13 adminis-

trative regions of KSA (independent variable). Multivariate Linear Regression analysis using the stepwise method with odd ratios was used to determine associations between the dependent and independent variables. The Variance Inflation Factor (VIF) was used to determine multicollineriaty among the independent variables. A conservative VIF score of 5 and above was set as a cutoff for performing sensitivity analysis to further address any existing multicollnearity among the independent variables [27–29].

A community mobility index was calculated by extracted maximum variance in a principal component analysis (PCA). The suitability of PCA was judged through Kaiser–Meyer–Olkin and Bartlett's tests. All results were two-sided, and $p < 0.05$ was considered significant. These analysis were performed in SPSS (ver. 24: Chicago, IL, USA).

### 3. Results

#### 3.1. Epidemiology of COVID-19 across Various Regions of KSA

As of December 26 2020, the total number of COVID-19 cases in Saudi Arabia was 360,690 cases. The total number of deaths was 6168 with a cumulative incidence rate of 105.41/10,000 population. The overall case fatality rate was 1.71%, with a most significant increase (4.31%; $p < 0.01$) in the Al Jouf region, followed by Jazan (3.78%; $p < 0.05$) and Northern Borders (3.55%; $p < 0.05$). The least but not statistically significant decrease (0.5%) was observed in the Madinah region. The Eastern region had the highest cumulative incidence rate of 170.5/10,000 population, followed by Madinah and Asir regions with incidence rates of 132.13 and 121.06 per 10,000 people, respectively.

Table 1 shows multivariate logistic regression findings in which case the fatality rate due to COVID-19 was almost three times higher in the Al-Jouf region (OR = 2.93; 95% CI: 1.29–6.64), two-half times higher in the Northern Border (OR = 2.50; 95% CI: 1.08–5.81), and just over two-half times higher in the Jazan regions (OR = 2.51; 95% CI: 1.09–5.79) when compared with Riyadh the capital.

**Table 1.** Characteristics and odds ratios of multivariate logistic regression analysis of association between regional mobility indices and case fatality rate of the 13 administrative regions.

| Regions | Population | No. of Cases | Cumulative Incidence per 10,000 | No. of Deaths | Case Fatality Rate (%) | *p* Value | OR (95%CI) |
|---|---|---|---|---|---|---|---|
| Riyadh | 8,660,885 | 74,449 | 85.96 | 1246 | 1.67 | 0.001 | Reference |
| Makkah | 9,033,491 | 87,897 | 97.3 | 2333 | 2.62 | | 1.67(0.69–4.08) [ns] |
| Madinah | 2,239,923 | 29,595 | 132.13 | 148 | 0.5 | | 0.38(1.00–1.43) [ns] |
| Qassim | 1,488,285 | 13,978 | 93.92 | 192 | 1.37 | | 0.90(0.32–2.50) [ns] |
| Eastern | 5,148,598 | 87,784 | 170.5 | 822 | 0.94 | | 0.99(0.73–2.68) [ns] |
| Asir | 2,308,329 | 27,944 | 121.06 | 427 | 1.53 | | 1.02(0.38–2.75) [ns] |
| Tabuk | 949,612 | 4981 | 52.45 | 83 | 1.67 | | 1.00(0.37–2.69) [ns] |
| Hail | 731,147 | 7291 | 99.72 | 126 | 1.73 | | 1.12(0.43–2.92) [ns] |
| Northern Border | 383,051 | 2453 | 64.04 | 87 | 3.55 | | 2.50(1.08–5.81) * |
| Jazan | 1,637,361 | 12,130 | 74.08 | 458 | 3.78 | | 2.51(1.09–5.79) * |
| Najran | 608,467 | 6474 | 106.4 | 66 | 1.02 | | 0.64(0.21–1.96) [ns] |
| Al Baha | 497,068 | 4460 | 89.73 | 65 | 1.46 | | 0.89(0.32–2.49) [ns] |
| Al Jouf | 531,952 | 1254 | 23.57 | 54 | 4.31 | | 2.93(1.29–6.64) ** |
| Total | 34,218,169 | 360,690 | 105.41 | 6168 | 1.71 | | - |

Degree of statistical significance: ns > 0.05; * < 0.05; ** < 0.01; OR: odds ratio; CI: confidence interval.

#### 3.2. Community Mobility Data Findings

Table 2 summarizes the descriptive statistics for the COVID-19 daily effective reproduction number ($R_t$) and the COVID-19 mobility indices for specified categories of locations from 2 March to 26 December 2020. Transit stations had the highest observed mobility reduction with an average of −53.85% ± 16.69%; residential mobility had the least reduction with an average of 13.29 ± 7.62%. The average effective reproduction number ($R_t$) was 2.9355 ± 7.44 ranging between 0.31–66.50.

**Table 2.** Summary statistics of COVID-19 mobility indices for specified categories of locations and effective reproduction number.

| Categories of Locations/COVID-19 $R_t$ | N * | Minimum (%) | Maximum (%) | Mean (%) | SD (%) | Median (%) |
|---|---|---|---|---|---|---|
| Retail and recreation | 295 | −84 | 1 | −31.44 | 19.70 | −22 |
| Grocery and pharmacy | 295 | −72 | 11 | −14.70 | 12.63 | −11 |
| Parks | 295 | −81 | 32 | −28.29 | 22.94 | −19 |
| Transit stations | 295 | −88 | -6 | −53.85 | 16.69 | −50 |
| Workplaces | 295 | −79 | 4 | −28.02 | 15.33 | −24 |
| Residential | 295 | 1 | 35 | 13.29 | 7.62 | 10 |
| CMI | 295 | −23 | 1 | 8.98 | 4.616 | 7.1 |
| R(t) | 282 | 0.31 | 66.50 | 2.9355 | 7.44 | 1.91 |

* Number of days; SD: standard deviation; CMI: community mobility indices.

### 3.2.1. Correlation between Community Mobility and COVID-19 Effective Reproduction Number

The correlation coefficients results show that the daily effective reproduction number R(t) was positively correlated with retail and recreation (r = 0.773; $p < 0.001$), grocery and pharmacy (r = 0.643; $p < 0.001$), parks (r = 0.335; $p < 0.001$), transit stations (r = 0.538; $p < 0.001$), and workplace (r = 0.657; $p < 0.001$) and was negatively correlated with residential mobility change (r = −0.810, $p < 0.01$), as shown in Table 3.

**Table 3.** Correlation between community mobility and COVID-19 effective reproduction number.

| Categories of Locations | N | $R_t$ Pearson Correlation | $p$ Value |
|---|---|---|---|
| Retail and recreation | 282 | 0.773 ** | <0.001 |
| Grocery and pharmacy | 282 | 0.643 ** | <0.001 |
| Parks | 282 | 0.335 ** | <0.001 |
| Transit stations | 282 | 0.538 ** | <0.001 |
| Workplaces | 282 | 0.657 ** | <0.001 |
| Residential | 282 | −0.810 ** | <0.001 |

Degree of statistical significance: ** <0.01.

### 3.2.2. Community Mobility Change as Predictors of Daily COVID-19 Effective Reproduction Number

Multiple linear regression was performed to investigate the effect of public and residential mobility changes on the COVID-19 effective reproduction number; with the removal of highly correlated predictor (retail and recreation) from the model, the Variance Inflation Factor (VIF) of the remaining predictors was less than five. Therefore, observed multicollinearity was not severe enough to necessitate further corrective measures. A significant regression equation was found (F (6, 275) = 19.69, $p < 0.001$), with an $R^2$ of 0.55. Therefore, $R_t$ = -0.533 + [0.685] (Grocery and pharmacy change) + [0.317] (Parks change) + [0.419] (Transit station change) + [0.658] (Work place change) + [−0.856] (Residential mobility change). Both mobility changes in public spaces and residential spaces were significant predictors of the COVID-19 effective reproduction number change with an exception of retail and recreation mobility change as shown in Table 4.

**Table 4.** Community mobility change as predictors of daily COVID-19 effective reproduction number.

| Categories of Locations | Coefficient (95% CI for Coefficient) | t | *p* Value | VIF |
|---|---|---|---|---|
| Constant | −0.533 (−9.375−−2.009) | −0.193 | 0.003 | - |
| Retail and Recreation | −0.079(−1.354−0.203) | −7.705 | 0.186 | 11.255 |
| Grocery and pharmacy | 0.685 (0.534−0.912) | 7.526 | <0.001 | 1.024 |
| Parks | 0.317 (0.191−0.546) | 2.384 | 0.001 | 2.594 |
| Transit station | 0.419 (0.138−0.977) | 7.839 | 0.025 | 1.866 |
| Work place | 0.658 (0.345−0.965) | 7.422 | 0.001 | 2.637 |
| Residential | −0.856 (−0.501−−0.987) | −8.028 | 0.001 | 3.434 |

### 3.2.3. Trend Analysis on Community Mobility Changes and Daily COVID-19 Effective Reproduction Number

There were indications that the implementation of mobility restrictions was associated with a relative sustained reduction in the COVID 19 effective reproduction number ($R_t$) from week three until the end of the study period. However, a more stable reduction in the effective reproduction number was observed from week eight and continued until the end of the study period (see Figure 1 below).

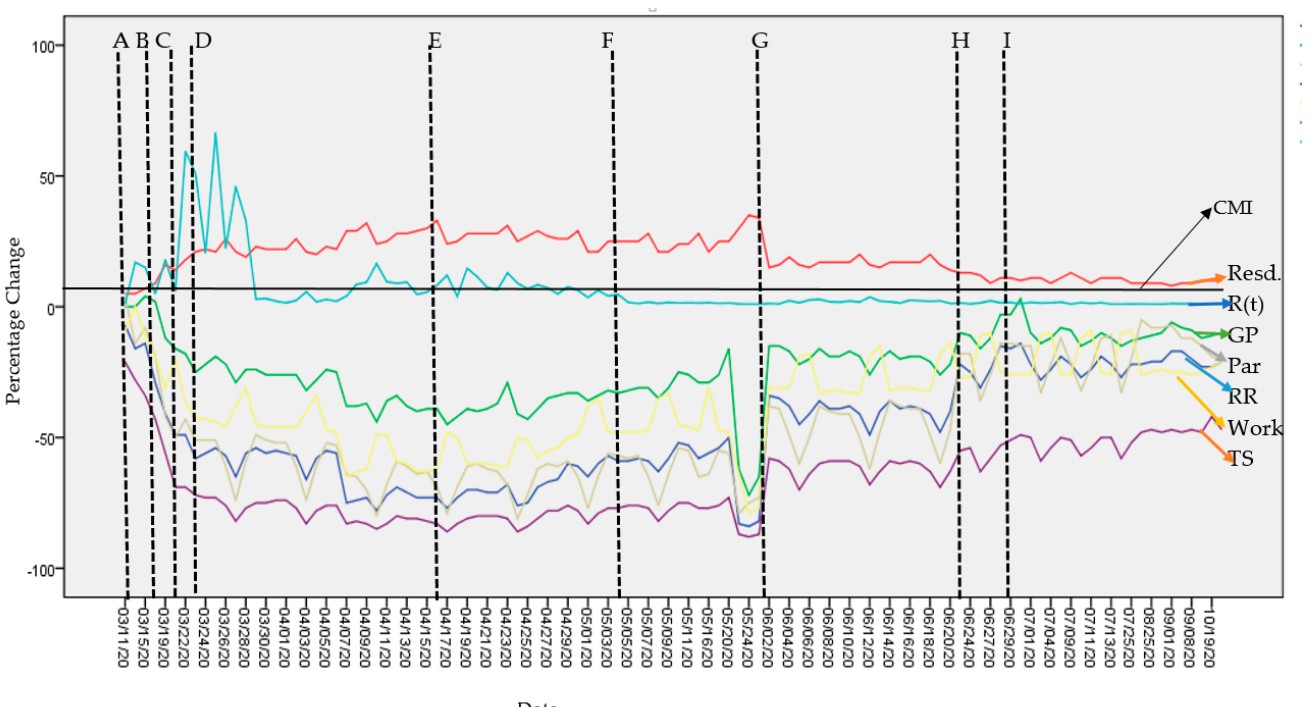

**Figure 1.** Time series analysis for daily mobility changes in Resd—Residential, GP—Grocery and Pharmacy, Par—Parks, RR—Retail and Recreation, Work—Workplaces, TS—Transit Stations, and R(t)—effective reproduction number. Note: The doted lines in Figure 1 indicate dates when certain KSA Government measures were initiated: A—WHO declared COVID-19 pandemic; B—Suspension of work place attendance and suspension of social events, sports, domestic flights, and daily and Friday prayers in mosques; suspension of all domestic public transport, flights, trains, buses and taxis; D—Partial curfew implemented; E—Mass PCR screening test; F—Complete curfew across KSA; G—Curfew lifting gradually, allowing domestic travel between regions and cities, opening some economic activities and allowing Friday and daily prayers while adhering to precautionary measures. Lifting suspension of attendance to workplaces in government and government agencies. Lifting suspension of domestic public transport, flights, trains, buses, and taxis; H—Back to normal life with preventive health instruction and social distance; I—Hajj permitted with restrictions.

## 4. Discussion

Ascertaining the effective reproduction number ($R_t$) related to NPI, such as mobility restriction in a pandemic situation, is a more objective approach of assessing the impact of the choice of NPI on case fatality associated with the pandemic. The consequence of ascertaining $R_t$ offers real-time situational awareness [25,26]. In this study, we quantified the effects of community mobility changes for every point in time on the case fatality rate using the effective reproduction number $R_t$, a time-dependent metric that changes dynamically in response to community mitigation strategies. This study demonstrated the effectiveness of a mobility restriction strategy that has the propensity to curb the spread of COVID-19 and fatality associated with COVID-19.

We found that mobility changes in public spaces and residential spaces were significant predictors of COVID-19 $R_t$ with an exception of retail and recreation mobility change. Reduced public mobility was associated with reduced COVID-19 $R_t$. Conversely, elevated residential mobility was associated with reduced COVID-19 $R_t$. Our findings are comparable with an analysis performed by [30] to determine the relationship between the daily number of COVID-19 cases and the use of public transport in the context of 2nd wave of the pandemic, which was more attributed to intra-regional movements in Italy. The pattern of their findings confirmed that new daily COVID-19 cases were directly associated to the movements through public transport, particularly in the regions with a high average usage of public transport. This was explainable as both a direct and indirect means of virus transmission during travels through public transport to contribute to the epidemic. The direct spread of the COVID-19 infection is due to over-crowding and congestion during travel in buses and vans; besides, people usually travel either for purpose of business or other reasons of social interaction, thus being an indirect source of spread. Mobility restriction in addition to the wearing of face masks and social distancing are non-pharmaceutical intervention (NPI) that have been demonstrated to effectively reduce COVID-19 transmission [6,31–33]. With the protracted nature of the COVID-19 pandemic, there is empirical evidence to justify the mandating and enforcing of NPI in addition to vaccination and other pharmacological interventions to further reduce case fatalities associated with COVID-19 [20].

A noticeable but fluctuating reduction in $R_t$ was observed from week three and continued until week eight before stabilizing. This finding may be attributed to an initial limited understanding of the natural history and data gathering related to SARS- CoV-2 and COVID-19. Furthermore, this limited understanding of the virus may have contributed to hesitancy associated with adoption of recommended NPIs. For example, in the study conducted by Al-Sharmmary (2021), only study participants who perceived government recommendation of NPIs to curb the spread of the virus as useful were about twice as likely to adhere to self-protective behaviors [16].

The findings of the trend analysis in this study aligned with the results of the prior ecological survey from Saudi Arabia conducted by Alyami and colleagues [10]. They reported that daily active cases continue to increase (determined by $R_t$ in our study) until the eighth week from the start of the pandemic when these active cases decreased and stabilized. This decrease in daily active cases was also associated with reduced case fatality associated with COVID-19 [10]. The decrees probably could be due to the KSA government having stronger enforcement of mobility restrictions in public areas (retail and recreation, grocery and pharmacy, parks, transit stations, and workplaces) compared to residential areas through the use of procedures in the "Tawakkalna" software application. The "Tawakkalna" software application is a nationwide-mandated application that monitors, records, and reports the current COVID-19 positivity and vaccination status of each resident. In addition, the software application is used for contact tracing to ensure social distancing and to determine admittance into public places [34].

The findings of our study further substantiate the effectiveness of NPI measures implemented by KSA in reducing the transmission of SARS-CoV-2 and its associated fatalities. Some of these measures include self-quarantine, self-isolation, contact tracing,

social distancing, school closures, travel suspension, crowd gathering reduction, and mobility restrictions [31]. Specifically, KSA suspended religious rite such as the Umra pilgrimage on 4 March 2020. Furthermore, on 23 March, mobility restriction by way of a partial curfew was implemented. The variations in regional distribution of COVID-19 CFR observed in the Al Jouf, Jazan, and Northern Border administrative regions ($\geq 2.5$ times higher CFR) compared to Riyadh may have been the result of the early implementation, enforcement, and adoption of travel restrictions implemented by the KSA government.

This study adds to a list of studies [6,7,35–39] examining the effectiveness of mobility restrictions in curbing the transmission of SARS-CoV-2. It is the first study in KSA that employed $R_t$ in assessing the effectiveness of the impact of mobility restrictions on the transmission of SARS-CoV-2.

This study was not without limitations. It was an observational and cross-sectional design without temporality. Due to varying regional mobile technology penetrations, the actual indices of mobility restriction may be underreported. Additionally, of those who own mobile devices, some individuals may choose not to turn on the location histories on their devices and that may have further underestimated the reported indices for mobility restrictions. As such, our findings may not be generalizable to the entire KSA population. Despite the limitations of this study, our analysis was robust enough to show that COVID-19 containment strategies that focus on reducing mobility in specified public locations such as grocery shops, pharmacies, parks, transit stations, and workplaces are effective in curbing transmission and fatalities associated with COVID-19.

## 5. Conclusions

The findings from this study contribute to prior evidence-based global research indicating that appropriately implemented non-pharmacologic interventions are potentially effective in stemming the spread and attendant fatalities associated with the COVID-19 pandemic. Early tracking and responses of stakeholders including government and decision makers in gathering data, understanding the natural history of a potential pandemic situation, clear and consistent messaging, and adequate allocation of resources can further enhance adherence of citizens to the recommended non-pharmacological intervention measures. These may further increase the impact of non-pharmacological intervention measures in curbing the spread of any potential pandemic.

**Author Contributions:** Conceptualization, M.A.A., C.O.A., S.A., N.E.E.E., A.K. and M.I.H.; data curation, A.K.; formal analysis, M.A.A., C.O.A., A.A.A. and R.A.Y.; funding acquisition, M.A.A.; methodology, A.A.A., R.A.Y., M.A.A., C.O.A., S.A. and S.-u.-N.H.; project administration, M.A.A.; software, A.A.A., R.A.Y., M.A.A. and A.K.; supervision, M.A.A.; writing—original draft, M.A.A., C.O.A., S.A., A.A.A., R.A.Y., N.E.E.E., A.K., S.-u.-N.H. and M.I.H.; writing—review and editing, M.A.A., C.O.A., N.E.E.E., A.K., S.-u.-N.H., M.I.H., A.A.A. and R.A.Y. All authors have read and agreed to the published version of the manuscript.

**Funding:** This research was funded by the Deanship of Scientific Research, University of Hail, Kingdom of Saudi Arabia, grant number COVID-1934.

**Institutional Review Board Statement:** The study was reviewed and approved as exempt by the Institutional Review Board of the University of Ha'il under project number COVID-1934 since this study did not involve humans or animals.

**Informed Consent Statement:** Not applicable.

**Data Availability Statement:** COVID-19 community mobility reports and epidemiological data were obtained from the Google website (https://www.google.com/covid19/mobility/ (accessed on 13 January 2022)) and KSA Ministry of Health website (Available online: https://covid19.moh.gov.sa/ (accessed on 13 January 2022)), respectively.

**Acknowledgments:** The authors would like to thank Scientific Research Deanship at the University of Hail, Saudi Arabia, for their research funding and wide contribution. Particular thanks to the Dean of College of Public Health and Health Informatics, University of Hail for continuous encouragement and support. Special thanks to Usman Atique for English editing of the manuscript.

**Conflicts of Interest:** The authors declare no conflict of interest.

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
