# Peer review of "Effectiveness of Human Mobility Change in Reducing the Spread of COVID-19: Ecological Study of Kingdom of Saudi Arabia"

_sustainability, doi:10.3390/su14063368_

Round 1

Reviewer 1 Report

This paper, certainly interesting, has two major flaws: lack in the Materials and Methods section of the description of 1) how the mobility of individuals is characterized and measured and 2) how the reproduction rate is defined and calculated. Moreover, the results section does not allow to get an idea of these aspects.
Therefore, I cannot evalute how appropriate is this work. I suggest to the authors to carefully address these issues before consideration for reviewing.

Author Response

Thank you for allowing us to submit a revised draft of our manuscript titled “Effectiveness of human mobility change on reducing the spread of COVID-19: Ecological study of Kingdom of Saudi Arabia” to MDPI Sustainability. We appreciate the time and effort you have dedicated to providing your valuable feedback on our manuscript. We have incorporated changes to reflect all of the suggestions used track changes within the manuscript.

Reviewer 2 Report

Thank you for giving me this opportunity to review this manuscript. 

Enclosed manuscript deals with very actual problem and from the western perspective it's even more interesting to find out how COVID19 has affected arab countries. Do we have similar conclusions, thoughts and results? I reviewed it with great interest, as it is current and vital.

In this particular case authors've investigated mandatory restrictions and changes in human mobility in the Kingdom of Saudi Arabia. The structure of the manuscript is clear and sounds scientifically. Authors have used relevant methods with sufficient explanations.The paper reflects a good choice of topic and a great deal of hard work.

However, I have several problems with the manuscript, mostly with latest and crucial literature and the last paragraph. To be more precise, I have these remarks on the quality of the paper:

1. In the line 48 Authors describe implementing non-pharmacologic interventions (NPI's) with 15 referenaces in the bracket. Most of them are about arab countries. On the one hand it shows good local literature overview, but on the other hand it is necessary to include international literature. Moreover, 15 referances in one bracket it's in my opinion not appropriate. I recommend to expand the explanations with specificity of every single referance. This part needs to be developed.
2. I recommend to include several sentences with an attempt to describe the thoughts from the line 211 "We found the mobility..." I'm curious what stands behind this in authors opinions?
3. In the same paragraph, between the lines 217-220, starting with phrase "With the protracted nature of the COVID19 pandemic..." please put the referances in the end of the sentence. When it's written that "there is empirical evidence" it's worth to confirm that with  publications.
4. It's generally worth to include more worldwide literature of presented topic. 
5. I propose developing the last paragraph (extend the description of possible possibilities to implement the research results).
6. I assume that it's acceptable in the first version, but number 13 and 14 from the reference list refers to the same position (reference is doubled)

Author Response

Thank you for allowing us to submit a revised draft of our manuscript titled “Effectiveness of human mobility change on reducing the spread of COVID-19: Ecological study of Kingdom of Saudi Arabia” to MDPI Sustainability. We appreciate the time and effort you have dedicated to providing your valuable feedback on our manuscript. We have incorporated changes to reflect all of the suggestions using track changes within the manuscript.

Reviewer 3 Report

Dear authors, dear editors,

the article is extremely relevant. The methodology and results are explicitly described.  The statistical results and limitations are clearly explained and discussed. I highly recommend the article for publication. 

The article has few typos (e.g., line 105, 197, 214, 234, 238).

Thank you for the opportunity to review this interesting article.

Yours sincerely

Dr. Ralf Hedel
Research Associate at Fraunhofer IVI (Germany, Dresden)

Author Response

(The authors gave the same response as above.)

Round 2

Reviewer 1 Report

Few more comments:

Line 123: « t »  should be « t_i »
Line 125, Eq.(1): Put the « i » as index
Lines 122 and 132:  the formulas used in Refs.[25] and [26] are different. 

Table 1: It is note clear which independent variables are used to provide the OR. Clearly state that by providing the expression like in line 208

Line 208: define what is « y »

Table 4:  define what is « B » (which should be « beta ») and relate that to line 208

Author Response

Thank you for allowing us to submit a revised draft of our manuscript. We appreciate the time and effort that the reviewer has dedicated to providing valuable feedback on our manuscript. We have been able to incorporate changes to reflect all of the suggestions provided by the reviewer. Here is a point-by-point response to the reviewers’ comments and concerns.
